# Coxsackievirus and adenovirus receptor expression facilitates enteroviral infections to drive the development of pancreatic cancer

Ligia I. Bastea[1,7], Xiang Liu[1,2,7], Alicia K. Fleming [1], Veethika Pandey[1], Heike Döppler[1], Brandy H. Edenfield[1], Murli Krishna[3], Lizhi Zhang[4], E. Aubrey Thompson [1], Paul M. Grandgenett[2], Michael A. Hollingsworth [2], DeLisa Fairweather [5], Dahn Clemens[6] & Peter Storz [1] ✉

The development of pancreatic cancer requires both, acquisition of an oncogenic mutation in *KRAS* as well as an inflammatory insult. However, the physiological causes for pancreatic inflammation are less defined. We show here that oncogenic KRas-expressing pre-neoplastic lesion cells upregulate coxsackievirus (CVB) and adenovirus receptor (CAR). This facilitates infections from enteroviruses such as CVB3, which can be detected in approximately 50% of pancreatic cancer patients. Moreover, using an animal model we show that a one-time pancreatic infection with CVB3 in control mice is transient, but in the presence of oncogenic KRas drives chronic inflammation and rapid development of pancreatic cancer. We further demonstrate that a knockout of CAR in pancreatic lesion cells blocks these CVB3-induced effects. Our data demonstrate that KRas-caused lesions promote the development of pancreatic cancer by enabling certain viral infections.

Premalignant pancreatic intraepithelial neoplasia (PanIN) and other ductal lesions are found in ~80% of people at age 70 or higher[1–3]. While most PanIN lesions do not progress to tumors in humans[4], pyrosequencing has shown that ~94–96% of these lesions have oncogenic mutations in *KRAS*[5,6]. However, virtually all pancreatic ductal adenocarcinoma (PDA) express oncogenic KRas, indicating that this oncogene is necessary but not sufficient for progression from premalignant lesions to pancreatic cancer.

In animal models, inflammation has been identified as a factor that accelerates the progression of KRas-related pre-neoplastic lesions by overcoming the senescence barrier and, therefore, favoring the development of PDA[7–9]. In humans, chronic pancreatitis is a risk factor for the development of PDA and can be caused by excessive alcohol use, metabolic disorders, smoking, gallstones, and genetic factors[10–12]. Inflammation resulting from microbial infections has been linked to several cancers. For example, *Helicobacter pylori* caused gastritis is associated with gastric cancer, papillomavirus-mediated chronic inflammation with cervical cancers, and hepatitis B and C viruses with liver cancers. However, the contribution of infections to the development and progression of pancreatic cancer is not well understood and a functional relationship between *KRAS* mutations and viral infections, as an etiological cause of human pancreatic cancer, has never been studied.

[1]Department of Cancer Biology, Mayo Clinic, Jacksonville, FL 32224, USA. [2]Eppley Institute for Research in Cancer and Allied Diseases, Fred & Pamela Buffett Cancer Center, University of Nebraska Medical Center, Omaha, NE 68198, USA. [3]Department of Laboratory Medicine and Pathology, Mayo Clinic, Jacksonville, FL 32224, USA. [4]Department of Laboratory Medicine & Pathology, Mayo Clinic, Rochester, MN 55905, USA. [5]Department of Cardiovascular Diseases, Mayo Clinic, 4500 San Pablo Road, Jacksonville, FL 32224, USA. [6]Department of Internal Medicine, Division of Gastroenterology and Hepatology, University of Nebraska Medical Center, Omaha, NE 68198, USA. [7]These authors contributed equally: Ligia I. Bastea, Xiang Liu. ✉e-mail: storz.peter@mayo.edu

Human infections with group B Coxsackie virus are a relatively common occurrence. Six serotypes CVB1-CVB6 have been identified so far[13]. They are a common cause of acute inflammation of the human pancreas and heart[14]. Here we show that CVB infection is a common occurrence in pancreatic cancer. Using CVB3 as an example, in a precancerous animal model, we demonstrate that the infection is facilitated by the expression of coxsackievirus and adenovirus receptor (CAR) in pancreatic lesions. Once present, CVB3 accelerates processes driving pancreatic cancer development by conserving an inflammatory environment and abrogating the senescence barrier in precancerous cells. These findings not only could alter our biological understanding of how pancreatic cancer develops, but also may impact the areas of early diagnostic techniques and targeted prevention therapies.

## Results and discussion

### PanIN cells upregulate coxsackievirus and adenovirus receptor

To determine factors that are upregulated when oncogenic KRas is expressed in PanIN cells and may mediate inflammation or increase the susceptibility to pancreatitis, we performed RNAseq analysis of microdissected PanIN lesions and adjacent acinar cells from p48[cre];LSL-Kras[G12D] (KC) mice. Comparison of the gene expression profiles identified *Cxadr*, a gene encoding coxsackievirus and adenovirus receptor (CAR), within a group of genes that were most significantly (log2FC ≥ 7) increased in PanIN as compared to acini (Fig. 1A). These data were confirmed using immunohistochemistry (IHC) staining for CAR protein in samples of KC or control mice where we detected CAR expression in PanIN, but not in adjacent acinar cells (Fig. 1B). CAR is a transmembrane receptor mainly expressed in epithelia and cardiac muscle, and its normal cellular function is to control cell polarity and tissue homeostasis[15]. However, CAR also serves as a receptor for group B coxsackieviruses and subgroup C adenoviruses (adenovirus 2 and 5)[16,17]. This function of CAR plays an important role in the pathogenesis of cardiac inflammation (myocarditis)[18], which can be caused by infection with CVB viruses[19].

To determine if CAR expression is upregulated during oncogenic KRAS-induced ductal metaplasia of acinar cells we isolated acini from LSL-Kras[G12D] mice, adenovirally infected them with GFP (control) or GFP;Cre to induce the expression of KRas[G12D], which drives their transdifferentiation (acinar-to-ductal metaplasia; ADM) to duct-like cells (Fig. 1C, bottom pictures). Acinar cells that underwent ADM to a PanIN cell-like phenotype showed an over 10-fold increase in the expression of CAR (Fig. 1C, bar graph). However, primary acinar cells that express KRas[G12D] do not induce CAR expression per se (Supplementary Fig. S1A), suggesting that additional (ADM-related) events are needed to drive CAR expression during KRAS-induced lesion formation.

### CAR (*CXADR*) expression is upregulated in human PDA

We next analyzed *CXADR* expression in normal and tumor samples from the TCGA TARGET GTEx dataset and found that its expression is upregulated in pancreatic ductal adenocarcinoma patient samples (Fig. 1D), correlating with decreased overall survival of patients (Supplementary Fig. S1B). Moreover, using tissue microarrays (TMAs) of patient tissue samples from two different sources (Mayo Clinic and US Biomax), we found that CAR expression was present in ~97% of human pancreatic cancers (Fig. 1E, F). CAR expression was already detected in low-grade lesions (ADM, PanIN1) adjacent to tumor, and in carcinoma in situ (PanIN3), but not in adjacent 'normal' acinar areas (Supplementary Fig. S1C).

### Incidence of CVB3 infection in pancreatic cancer patients

Infection with CVB induces inflammatory cytokine expression and immune responses[20,21]. Although in some studies CVB3 has been associated with ~40% of cases of acute and chronic pancreatitis[14,22], the role of CVB3 replication and infection-mediated inflammation in the development of pancreatic cancer has never been examined. We, therefore, determined whether CVB3 could be detected in clinical samples of pancreatic adenocarcinoma (Fig. 2A). We detected CVB3 in 53% of surgically resected patient samples from Mayo Clinic (*n* = 49), 45% of surgical resection samples from a commercial source (US Biomax, *n* = 60) and 51% of surgical resection samples from UNMC (*n* = 37) (Fig. 2B). In contrast to pancreatic cancer, only 1 out of 125 breast cancer biopsies showed evidence of CVB3 (Fig. 2B). This is not surprising because oncogenic KRas mutations are marginal in breast cancer and CVB3 is typically a gastrointestinal infection. However, it is possible that enteroviral infections contribute to other KRas-initiated cancers. For example, oncogenic *KRAS* mutations are known to occur in about 25% of lung adenocarcinomas[23], and CVB3 can also infect lung adenocarcinoma cells[24].

To confirm our immunohistochemistry data, we performed in situ hybridization (ISH) for viral RNA and observed a similar distribution of CVB3-infected pancreatic patient samples (Fig. 2C, Supplementary Fig. S1D). This finding was confirmed with a second ISH probe (Fig. 2D, Supplementary Fig. S1E), and the specificity of the ISH probe for CVB3 was determined by analyzing samples of mouse pancreata that were infected with CVB3, CVB4, or CVB5 (Supplementary Fig. S2A). Moreover, a proximity ligation assay (PLA) demonstrated that CVB3 and CAR interact in PanIN lesions (Fig. 2E; red dots indicate CVB3:CAR interaction). It should be noted that CVB3 is not the only coxsackievirus serotype that can be found in the pancreata of PDA patients, and we detected the presence of all 5 other coxsackie serotypes (Supplementary Fig. S2B).

### A single CVB3 infection can rapidly lead to pancreatic cancer

To determine if the presence of CVB in patient samples was due to a secondary infection or if it could be a causative factor for the development of PDA, we utilized KC mice as a model system. To test whether CVB3 infection contributes to the development of PDA, KC mice were (one-time) infected with CVB3 at 8 weeks of age (Fig. 3A). 6 weeks after infection pancreata were harvested and analyzed. At this stage antibodies against CVB3 were detectable in the sera (Supplementary Fig. S3A) and RNA for virus capsid in pancreata of all CVB3-infected mice (Supplementary Fig. S3B). In non-transgenic control mice, infection with CVB3 led to acute pancreatitis, which at the endpoint of the experiment was resolved (Fig. 3B, bottom row). In the presence of oncogenic KRas[G12D], CVB3 infection led to a dramatic increase in pancreatic abnormal structures, as compared to KRas[G12D] alone (Fig. 3B, top row). This increase in abnormal structures was similar to experimental models where KC mice are treated with caerulein to induce inflammation or when additional p53 mutations are induced (KPC model) to drive tumor development and progression (Supplementary Fig. S3C).

At the endpoint, pancreata of control mice treated with CVB3 had a mostly regenerated pancreas, with a 29 ± 13% increase in fat tissue (Fig. 3C), possibly due to pancreatitis-induced acinar cell death and replacement by adipocytes[25,26]. KC mice at the endpoint of the experiments showed ~17 ± 8% abnormal structures (early precancerous lesions and fibrotic areas), while the additional CVB3 infection increased abnormal structures to 77 ± 5%. Moreover, there was a significant increase in precancerous lesions between KC & vehicle and KC & CVB3-infected mice (ADM, PanIN1A/B, PanIN2—all *p* < 0.0001). In addition, lesions were more progressed as evidenced by a significant increase in PanIN2 and the occurrence of PanIN3 (carcinoma in situ) and areas of PDA in KC mice infected with CVB3 (Fig. 3D). Supplementary Fig. S3D shows some of the progressed areas, and staining for cytokeratin-19 (CK-19) indicated cells disseminating into the stroma (asterisk). Individual analyses of mice per group did not indicate an obvious effect of sex on CVB3-induced lesion progression (Supplementary Fig. S4).

### CVB3 infection drives inflammation-related events

For the CVB serotype CVB4, it has been shown that infection of pancreatic cells leads to changes in host and virus[27]. Similarly, CVB3 may contribute to pancreatic cancer progression by multiple mechanisms.

For example, CVB3 can drive pancreatic inflammation and fibrosis[28]. This is mediated by host induction of antiviral IFNs and pro-inflammatory mediators such as TNF, IL-6, IL-8, MIP1α, and MIP1β[29]. COX-2 expression is a marker for inflammatory processes and a marker

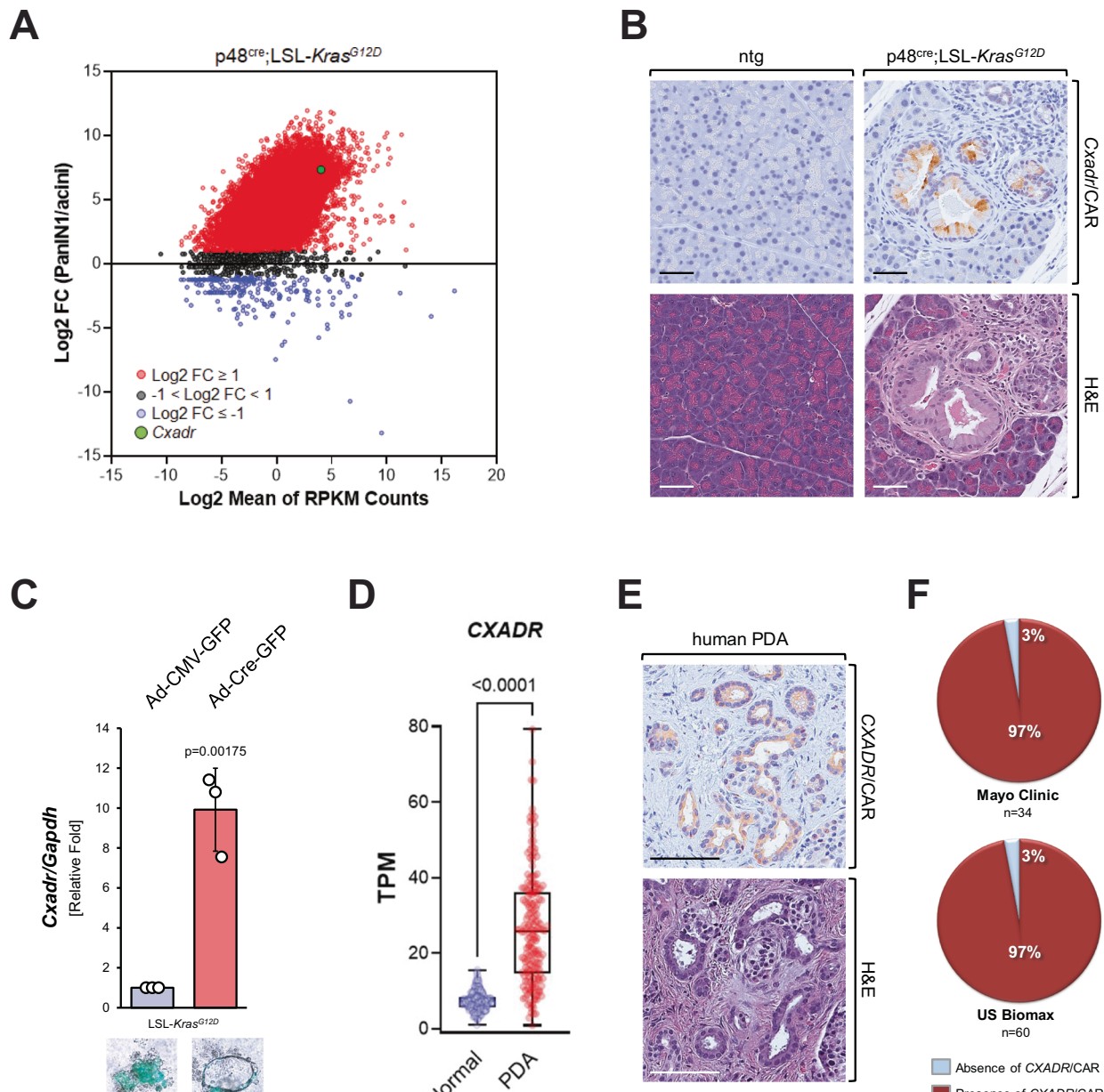

**Fig. 1 | Oncogenic KRas-expressing mouse PanIN cells upregulate CAR, which is expressed in human PDA. A** PanIN1 and acini were microdissected from KC (p48[cre];LSL-Kras[G12D]) mice and subjected to RNA-seq analyses. Shown is a graphical representation of the relative abundance of transcripts. RNA-seq data were averaged from 40 laser-dissected PanIN1 lesions and an equivalent area with normal acini. Data were filtered to exclude transcripts with a sum RPKM < 0.5 so as to eliminate transcripts at/or below the limits of detection, and 20,037 transcripts remained after filtering. Shown is a plot of the ratio of the averages of the individual transcripts in both sample types, expressed as log2-fold change (FC) PanIN/acini. Black dots represent log2 FC < 3, red dots 3 ≤log2 FC < 7 and blue dots log2 FC ≥ 7. The green dot indicates *Cxadr*/CAR. Source data are provided as a Source Data file. RNAseq raw data can be accessed in Gene Expression Omnibus (GEO) using accession code GEO:GSE280352. **B** Pancreatic tissues of p48[cre];LSL-Kras[G12D] and ntg control mice were analyzed by IHC for expression of CAR (brown) in acinar cell regions or PanIN lesions. The bottom pictures show H&E staining of a serial section. Shown are representative pictures of *n* = 4 biological replicates (ntg or KC mice). **C** Primary acini were isolated

from an LSL-Kras[G12D] mouse and infected with adenovirus harboring GFP (control) or GFP and cre (GFP;cre). Cells were embedded in collagen (3D culture). After 5 days, cells/structures were photographed to show GFP expression (bottom, representative pictures, scale bar indicates 50 μm) and then harvested and subjected to quantitative RT-PCR analysis for *Cxadr*/CAR and GAPDH. The bar graph shows data from *n* = 3 biological replicates. Data are presented as mean values ± SD. Statistical significance (*p* = 0.00175) was determined using a two-sided *t*-test. Source data are provided as a Source Data file. **D** *CXADR* expression in normal (*n* = 195) and tumor (*n* = 176) samples from the TCGA TARGET GTEx dataset where the midline of the box represents the median. The whiskers extend to min and max. Outliers were removed via ROUT (*Q* = 1%) and statistical significance (*p* < 0.0001) was determined using a two-sided *t*-test. Source data are provided as a Source Data file. **E, F** Tissue microarrays containing human PDA samples (US Biomax *n* = 60 samples; Mayo Clinic *n* = 34 samples) were analyzed by IHC for expression of CAR. **E** shows an example of an IHC for CAR (brown, top) and H&E staining of a serial section (bottom). **F** shows the quantification of CAR expression in both TMAs. Source data are provided as a Source Data file.

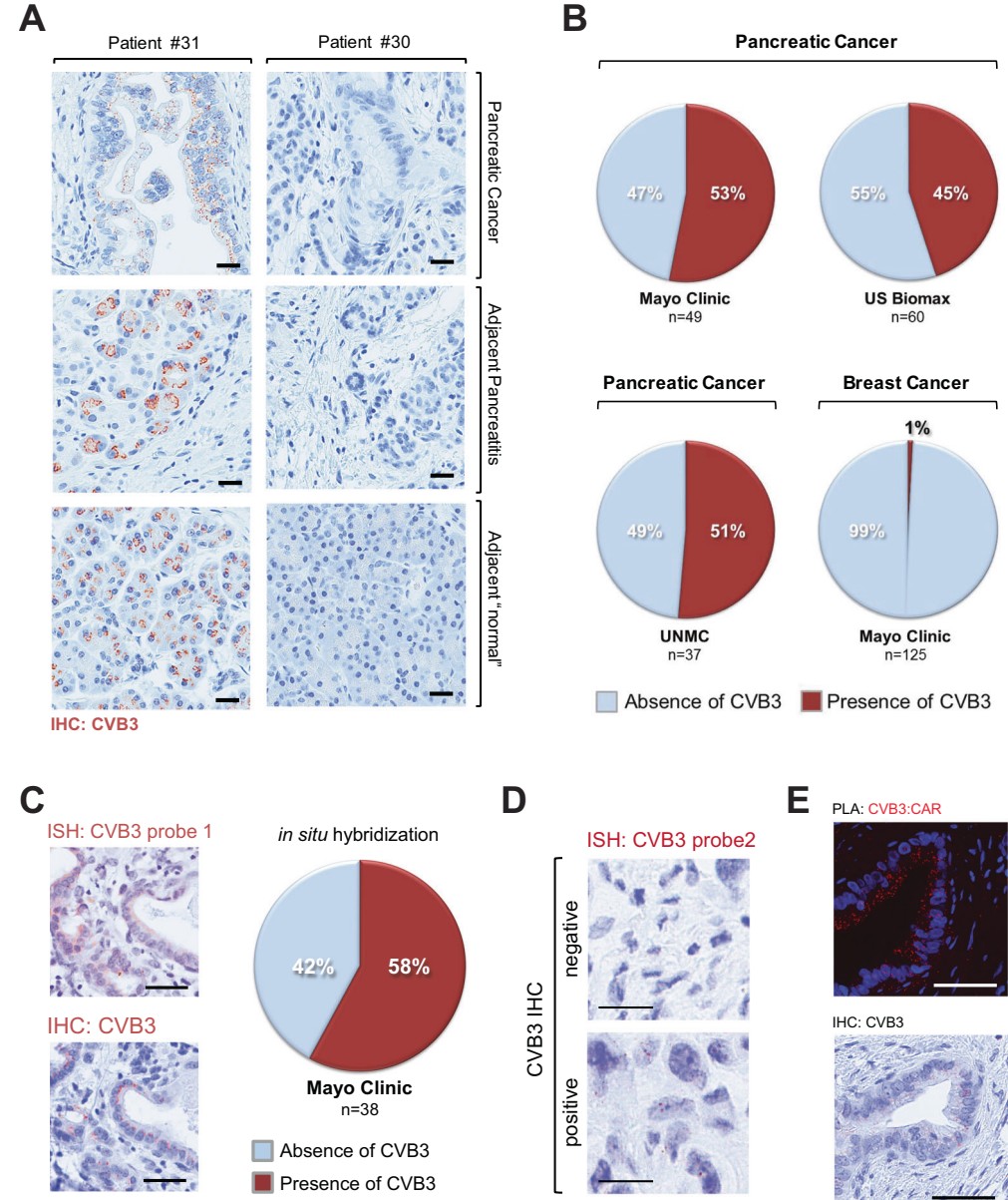

**Fig. 2 | Incidence of coxsackievirus B3 infection in pancreatic cancer patients.** **A** Tissue samples from human pancreatic cancer patients ($n = 49$) were analyzed by IHC for the presence of CVB3. Shown are areas of pancreatic cancer, adjacent pancreatitis, and "normal" acinar areas from tumors of the same patient. The left side row shows a patient positive for CVB3 (brown dots) and the right side row a patient negative for CVB3. **B** Tissue microarrays from different sources containing human PDA samples (US Biomax $n = 60$ samples; Mayo Clinic $n = 49$ samples, UNMC $n = 37$ samples) or breast cancer samples (Mayo Clinic $n = 125$ samples) were analyzed by IHC for the presence of CVB3. Quantifications show a % absence or presence of CVB3. Source data are provided as a Source Data file. **C** Serial sections of TMAs containing human PDA samples (Mayo Clinic $n = 38$ samples) were subjected to ISH probe 1 (top picture) or IHC (bottom picture) for CVB3. The scale bar indicates 50 μm. The pie graph shows the % presence or absence of CVB3 for the in situ hybridization data. Source data are provided as a Source Data file. **D** Human samples that were either positive or negative for CVB3 immunohistochemistry were analyzed with a second ISH probe (CVB3 probe2). Brown dots indicate CVB3 RNA. The scale bar indicates 10 μm. **E** Shows one of 3 biological repeats of a proximity ligation assay (PLA) in which samples were stained for the interaction of CVB3 and CAR. Red dots show the interaction (CVB3:CAR) of both molecules. The bottom picture shows an IHC for CVB3 in a serial section. The scale bar indicates 50 μm.

for abnormal expression of p53 in pancreatic cancer[30]. Our data now implicate that a one-time viral infection in the presence of oncogenic KRas increases the expression of COX-2 (Fig. 3E). In the pancreas, aberrant expression of COX-2 alone can induce the formation of pancreatic lesions[31], and in the presence of KRas[G12D] COX-2 accelerates the progression of PanIN[32]. This is because activation of COX-2 can lead to a positive feed-forward loop enhancing activities of oncogenic and wildtype KRas, which is needed for the development of pancreatic cancer[33,34]. Consequently, upregulation of COX-2 expression can be

detected in human pancreatic cancer and correlates with poor prognosis[35,36].

In line with the observed increase in inflammation and progression of pancreatic lesions from KC mice infected with CVB3, we detected an increased presence of neutrophils, T cells, and macrophages (Fig. 3F and Supplementary Fig. S5). T cells were mostly CD4+ (Supplementary Fig. S6A, S6B), which is in line with previously published work suggesting a driving role of CD4+ T cell populations in PDA by secreting immunosuppressive and fibrinogenic cytokines[37,38]. In

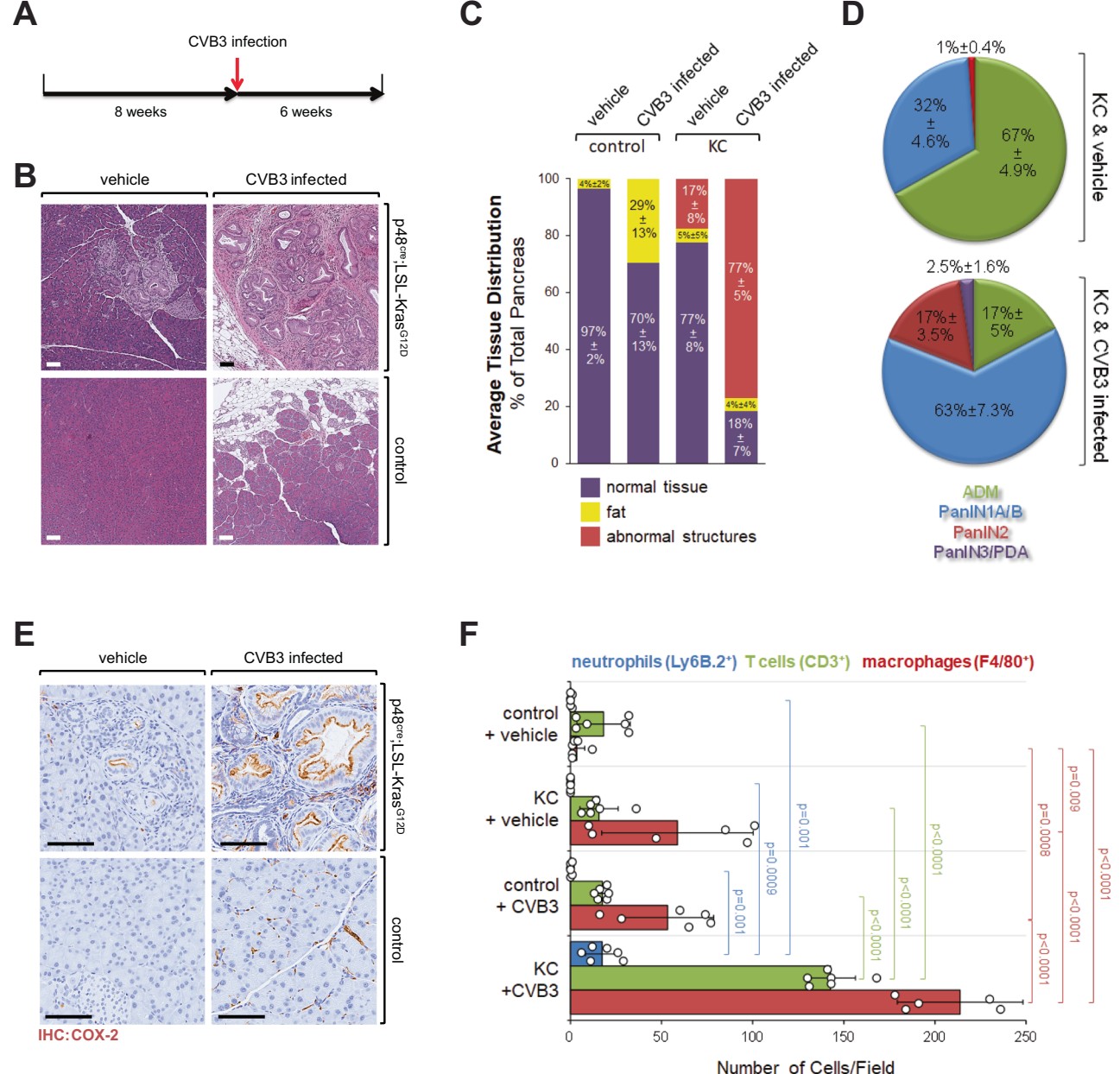

**Fig. 3 | A single CVB3 infection in the presence of KRas^G12D rapidly leads to pancreatic cancer.** **A** Scheme of the animal experiment. Mice at 8 weeks of age were infected with CVB3 once. After 6 weeks, pancreata were harvested and analyzed. **B** Pancreatic tissues of KC or control mice, either vehicle-treated or infected with CVB3, were stained with H&E. Pictures show a representative area of the pancreas. The bar indicates 100 μm. **C** Quantification of normal pancreas area, fat area, and areas with abnormal structures of $n = 5$ mice per group. Abnormal Structures: 3 vs. 4, $p < 0.0001$. Normal Tissue: 1 vs. 2, $p = 0.0025$; 1 vs. 3, $p = 0.0006$; 1 vs. 4, $p < 0.0001$; 2 vs. 4, $p < 0.0001$; 3 vs. 4, $p < 0.0001$; others: ns. Fat: 1 vs. 2, $p = 0.0025$; 2 vs. 3, $p = 0.0053$; 2 vs. 4, $p = 0.0038$; others = ns. Statistical significance between the two groups was determined using a two-sided $t$-test. Source data are provided as a Source Data file. **D** Quantification of ADM, PanIN1A/B, PanIN2, and PanIN3/PDA in KC & vehicle *versus* KC & CVB3 mice ($n = 5$ mice per group; see Supplementary Fig. S4 for individual mice and sex of mice). ADM $p < 0.0001$; PanIN1A/B $p < 0.0001$; PanIN2 $p < 0.0001$; PanIN3/PDA $p = 0.0088$. Statistical significance between the two groups was determined using a two-sided $t$-test. Source data are provided as a Source Data file. **E** Pancreatic tissues of KC or control mice, either vehicle treated or infected with CVB3, were stained by IHC for COX-2 expression. Shown are representative micrographs of $n = 4$ mice per group. The bar indicates 100 μm. **F** Pancreatic tissues of KC or control mice either vehicle treated or infected with CVB3 were stained by IHC for expression of Ly6B.2, CD3 or F4/80. The bar graph shows a quantification of $n = 6$ samples (individual data points shown as dots) per indicated group (bars: blue−neutrophils, green−T cells, red−macrophages). Data are presented as mean values ± SD. Statistical significance between two groups was determined using a two-sided $t$-test and $p$ values are indicated in the bar graph. Source data are provided as a Source Data file.

addition, we observed an increase in fibrosis as indicated by trichrome staining (Supplementary Fig. S6C), the presence of smooth muscle actin (SMA) expressing fibroblasts (Supplementary Fig. S6D) and the presence of fibrosis-organizing M2 (Ym1+) macrophages (Supplementary Fig. S7A, S7B).

## Deletion of CAR blocks CVB3 infection and progression to PDA

To demonstrate that CVB3 mediates these effects through CAR, we first generated mice with an acinar cell-specific CAR knockout (p48^cre;CAR^−/−). The knockout of CAR did not affect the normal morphology of the mouse pancreas (Supplementary Fig. S8A). In the

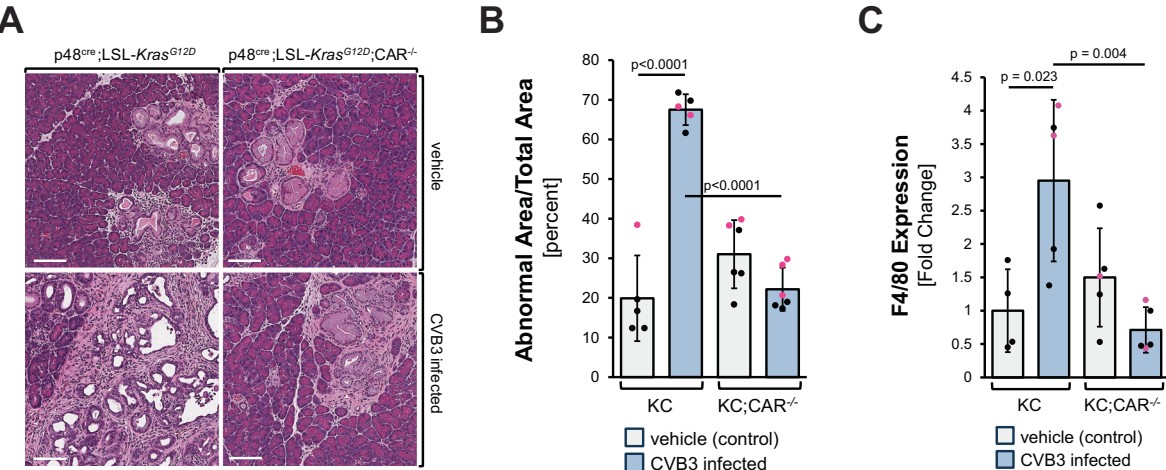

**Fig. 4 | Acinar cell-specific deletion of CAR blocks progression. A** KC or KC;CAR$^{-/-}$ mice at 8 weeks of age were infected with CVB3 once. After 6 weeks, pancreata were harvested and stained with H&E. Pictures show a representative area of the pancreas. The bar indicates 100 μm. **B** The bar graph shows analyses of the abnormal (fibrotic lesion) areas of pancreata of $n = 5$ or 6 mice per group (as indicated by the dots). Data are presented as mean values ± SD. Statistical significance between two groups was determined using a two-sided *t*-test and *p*-values are included in the graph. Sex of animals is indicated by the dot color (female−red; male−black). Source data are provided as a Source Data file. **C** Pancreatic tissues of KC or KC; CAR$^{-/-}$ mice either vehicle-treated or infected with CVB3 were analyzed for the presence of macrophages (F4/80). The bar graph shows analyses of the abnormal (fibrotic lesion) areas of pancreata of $n = 4$ (vehicle control) or $n = 5$ mice (other groups) per group (as indicated by the dots). Data are presented as mean values ± SD. Statistical significance between the two groups was determined using a two-sided *t*-test, and *p*-values are included in the graph. The sex of animals is indicated by the dot color (female−red; male−black). Source data are provided as a Source Data file. An accompanying representative IHC is shown in Supplementary Fig. S8C.

presence of oncogenic Kras, acinar cell-specific deletion of CAR (KC;CAR$^{-/-}$) did not affect the formation of pancreatic lesions (Fig. 4A, top row) but blocked progression to PDA after a one-time CVB3 infection (Fig. 4A, bottom row). This is confirmed by quantification of the abnormal pancreatic area between samples (Fig. 4B). Control IHC for CVB3 confirmed that a knockout of CAR blocked CVB3 infection (Supplementary Fig. S8B). Further analyses of pancreata from these mice confirmed that the deletion of CAR decreases the presence of macrophages (Fig. 4C; Supplementary Fig. S8C) and decreases the fibrosis-related presence of SMA+ fibroblasts (Supplementary Fig. S8D, S8E).

Thus, our data suggest that infection with CVB3 facilitated through CAR expression in oncogenic KRas-caused lesions contributes to the development of pancreatic cancer by promoting chronic inflammation and fibrosis. Our data also suggest that rather than being a cause for KRas mutations, inflammation is a consequence of KRas-caused lesion formation. The presence of inflammatory cells like macrophages not only increases fibrosis and lesion progression[39,40] but also may abrogate the KRas-induced senescence barrier in precancerous lesions and, therefore favor and accelerate the development of PDA[7,8].

### Decrease of the senescence barrier and rapid progression to PDA

As predicted by the increased inflammatory response caused by CVB3 infection in KC mice, we observed a decrease in senescence of PanIN cells as measured by detection of β-Galactosidase positive cells (Fig. 5A, a larger overview area is shown in Supplementary Fig. S9A, S9B) and by detection of p16$^{Ink4}$ positive cells (Fig. 5B, Supplementary Fig. S10A). Besides a loss of senescence, lesions in CVB3-infected KC mice showed an increase in proliferation as indicated by the presence of Ki67, and this was blocked when CAR was deleted (Fig. 5C, Supplementary Fig. S10B).

At 38 weeks abnormal areas of control-treated KC mice mostly showed low-grade lesions and desmoplasia, whereas one-time CVB3-infected KC mice showed progressed tumors, and in a few cases metastases to the liver (Fig. 6A). Eventually, we followed the survival of mice ($n = 7$ per experimental group) and found that KC mice that were one-time infected with CVB3 died of pancreatic cancer within 23 and 94 weeks, whereas KC control mice showed a significant ($p = 0.0471$) increase in survival and died within 58 to 118 weeks (Fig. 6B). Of note, within these experimental groups female or male animals were randomly assigned (indicated by symbols). While in some CVB3-induced myocarditis studies, male mice may have a slightly increased susceptibility to infection and severity of the effects[41], we did not observe any obvious effects on lesion progression (Supplementary Fig. S4, Fig. 4B) and survival (Fig. 6B) due to sex of the animal. However, to determine if CVB3 in KC mice could have effects based on sex, animal numbers would need to be significantly increased in future studies.

The rapid progression of KC mice to pancreatic cancer in the presence of a CVB3 infection may also be attributed to other factors than inflammation. For example, CVB3 has been shown to initiate p53 degradation[42] and to induce autophagy[43], which are both required for pancreatic cancer tumor development and progression[44]. The crosstalk between CVB3 infection and *KRAS* mutations in these processes needs to be addressed in future studies.

In summary, our analyses of 146 pancreatic cancer patient samples from different sources indicated that CVB3 infection is a common occurrence (in approximately 50% of all samples). Animal experiments indicated that the infection is not secondary, but can be facilitated by oncogenic KRas, and can accelerate and amplify processes driving PDA development by promoting a pro-inflammatory environment and abrogating the senescence barrier in PanIN cells (Fig. 6C). These findings could radically alter our biological understanding of how pancreatic cancer develops and improve early diagnostic techniques and targeted prevention therapies.

## Methods

Research in this paper complies with all relevant ethical regulations. All animal experiments were approved by the Mayo Clinic Institutional Animal Care and Use Committee (IACUC) and were performed in accordance with relevant institutional and national guidelines and

## A

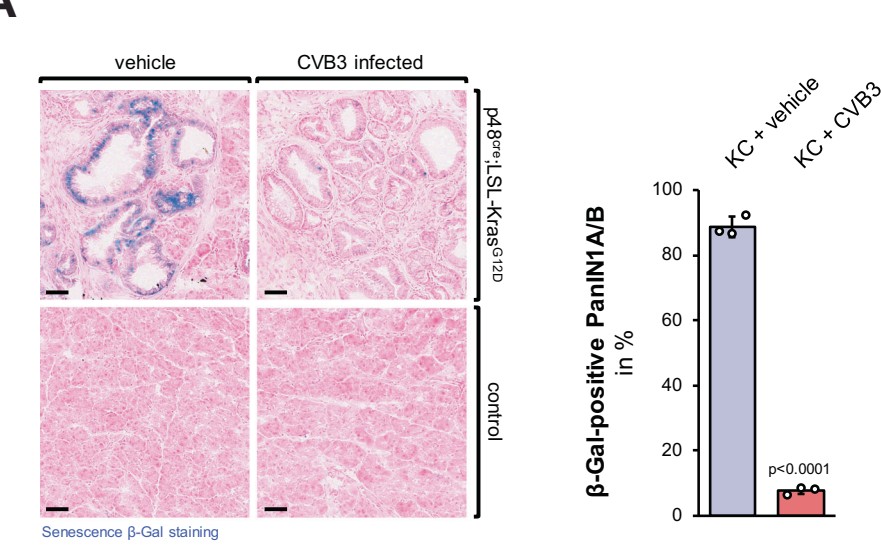

Senescence β-Gal staining

## B

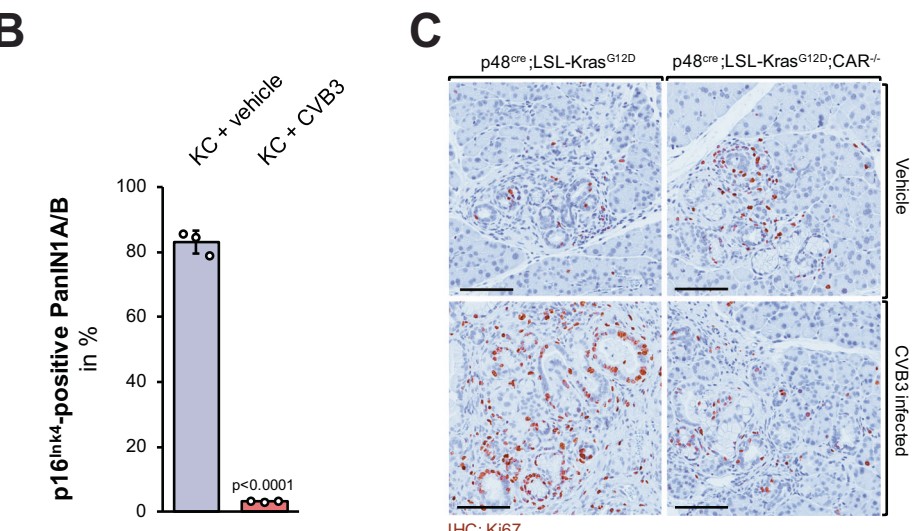

## C

IHC: Ki67

**Fig. 5 | CVB3 infection of KC mice leads to a decrease of the senescence barrier and increased proliferation of lesion cells. A** Pancreatic tissues of KC or control mice, either control-treated or infected with CVB3, were stained with the Senescence β-Gal Staining kit (Cell Signaling) and then counterstained with Nuclear Fast Red. The bar indicates 50 μm. The bar graph shows a quantification of $n = 3$ individual samples per indicated group. Data are presented as mean values ± SD. $p < 0.0001$ (two-sided $t$-test) indicates statistical significance. Source data are provided as a Source Data file. **B** Pancreatic tissues of KC or control mice either vehicle-treated or infected with CVB3, were analyzed by IHC for the presence of

p16$^{Ink4}$ (CDKN2). The bar graph shows a quantification of $n = 3$ individual samples per indicated group. Data are presented as mean values ± SD. $p < 0.0001$ (two-sided $t$-test) indicates statistical significance as compared to the vehicle-treated control mice. Source data are provided as a Source Data file. **C** Pancreatic tissues of KC or KC; CAR$^{-/-}$ mice either vehicle-treated or infected with CVB3, were stained pro-liferating cells (IHC for Ki67). Shown are representative pictures of lesion areas. The bar indicates 100 μm. The corresponding analysis is shown in Supplementary Fig. S10B.

regulations. Experiments were conducted under Mayo Clinic IACUC protocols A00001701-16-R19, A00006044-21-R24, A43615-15, and A00003891-18-R24. Research involving unidentifiable/de-identified biological specimens is not considered human-subject research. Staining of human tissue samples was approved by the Mayo Clinic Institutional Review Board (IRB) under protocols 14-009775, 18-005727, 19-001585, and 19-012566.

### Antibodies and reagents
CVB1 (Millipore; MAB944); CVB2 (Millipore; MAB946); CVB3 (Millipore; MAB948); CVB4 (Millipore; MAB941); CVB5 (Millipore; MAB943); CVB6 (Millipore; MAB945); CAR (Abcam; ab189216); CK-19 (Santa

Cruz; SC-33111); COX-2 (Cayman Chemical; 160126); CD3 (Abcam; ab5690); CD4 (Abcam; ab183685); CD8 (Abcam; ab209775); p16INK4 (Abcam; ab189034); DCLK1 (Abcam; ab37994); F4/80 (AbD Serotec; MCA497R); Ly6B2 (AbD Serotec; MCA771G); SMA (Abcam; ab5694); Ki67 (Abcam; ab15580); Ym1 (Stemcell Technologies; 60130). Supplementary Table 1 indicates dilutions for IHC and PLA. The Trichome Stain Kit and DAPI were from Sigma-Aldrich (St. Louis, MO). Human coxsackievirus B3 was from ATCC (Manassas, VA).

### Mouse lines, treatment and CVB3 infection
LSL-Kras$^{G12D}$ mice (obtained from the NCI Mouse Repository; MMHCC) were crossed with p48$^{cre}$ mice (a gift from Dr. Pinku

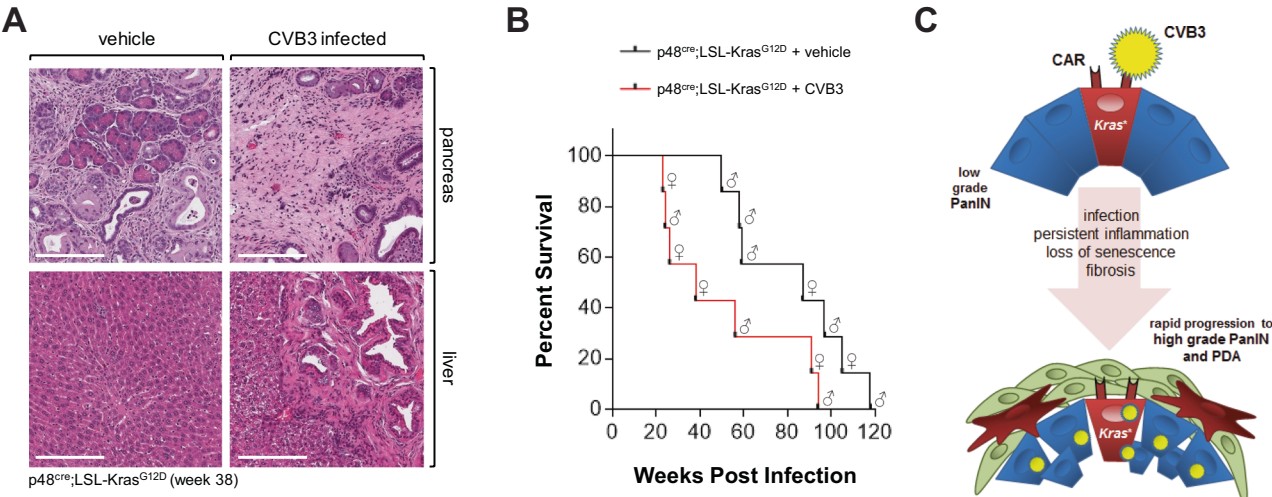

**Fig. 6 | CVB3 infection of KC mice leads to rapid progression to PDA. A** H&E staining of representative pancreas and liver tissue from (*n* = 4) control or CVB3 infected p48^cre^;LSL-Kras^G12D^ mice. Tissue was collected at week 38 after infection. The bar indicates 200 μm. **B** Kaplan–Meier curve showing significantly reduced survival in *n* = 7 CVB3-infected p48^cre^;LSL-Kras^G12D^ mice (red line) as compared to *n* = 7 PBS control p48^cre^;LSL-Kras^G12D^ (black line, *p* = 0.0471; Log-rank (Mantel–Cox) test. The median survival of KC + vehicle is 87 weeks, and the median survival of KC + CVB3 is 38 weeks. Sex of animals is indicated by the symbols. Source data are provided as a Source Data file. **C** Model of how CAR in oncogenic KRas expressing lesions facilitates CVB3 infection, and how this can contribute to the progression to high-grade PanIN and PDA.

Mukherjee, University of North Carolina) to generate bi-transgenic p48^cre^;LSL-Kras^G12D^ (KC) mice, C57BL/6J background[40,45]. *Cxadr^tm1.1lcs^*/J mice (strain #017359, B6;129S2 background) were obtained from The Jackson Laboratory (Bar Harbor, ME). LSL-p53^R172H^ and Pdx1^cre/+^ mice (both 129/SvJae/C57Bl/6 background) were a gift from Dr. Howard Crawford (Henry Ford Health)[40,45]. The mice were housed and bred in ventilated cages in a temperature and humidity-controlled barrier facility at Mayo Clinic, under a 12-h dark/light cycle. Food and water were provided ad libitum. For caerulein injections, mice were intra-peritoneally injected with caerulein (75 μg/kg body weight) once per hour for a time span of eight hours, two days in a row. For CVB3 infections, 8-week-old mice were intraperitoneally (IP) injected once with CVB3 ($6.22 \times 10^4$ PFU/0.1 mL) in a volume of 100 μL. For all animal survival studies we adhered to the endpoint criteria permitted by the institutional policy of the Mayo Clinic IACUC committee. These include tumors reaching ≥10% of body weight or weight loss ≥20% of body weight. Sex of animals was not considered in the study since pancreatic cancer development (lesion formation) in KC mice is not sex-dependent. We indicated the sex of animals in key experiments to demonstrate that observed CVB3 effects are also independent of sex in our model.

### Human pancreatic tissue samples

TMAs were sourced either from US Biomax Inc. in Rockville, MD, or produced from formalin-fixed paraffin-embedded (FFPE) patient tissues from archived materials at Mayo Clinic and UNMC, adhering to previously approved institutional review board (IRB) protocols and institutional guidelines. Rapid autopsy samples were from decedents who had previously been diagnosed with pancreatic ductal adeno-carcinoma and were obtained from the University of Nebraska Medical Center's Tissue Bank through their Rapid Autopsy Pancreatic (RAP) program. The specimens were collected within three hours of death and immediately frozen in liquid nitrogen or fixed in formalin. All samples analyzed were deidentified and are not considered human subject research.

### Microdissection

Tissue was embedded in Tissue-Tek OCT compound and frozen immediately after harvesting. Pancreas cryosections (10 μm) were stained with cresyl violet according to the LCM Staining Kit (Life Technologies, Carlsbad, CA). Briefly, PEN-Membrane 2.0 μm slides (Leica, Buffalo Grove, IL) were rinsed with RNase Zap and treated with UV light overnight. Cryosections were rinsed with alcohol (95%, 75%, and 50%), stained with cresyl violet for 20 sec, dehydrated in alcohol and xylene (50%, 75%, 95%, 100%, and xylene), and dried completely in a desiccator for 5 min. Tissues were microdissected within one hour of staining using a Leica LCM6500 microscope, collected in microtubes containing 30 μL RLT Lysis Buffer (RNeasy Micro Kit, Qiagen) and stored at −80 °C.

### Isolation of primary pancreatic acinar cells, infection, and 3D culture

The protocol for the isolation of primary pancreatic acinar cells has been described previously in detail[46]. To summarize, the pancreas was harvested, washed in ice-cold HBSS, cut, and digested with collagenase I in a shaker at 37 °C. The digested pancreas was washed twice with HBSS + 5% FBS and passed through 500 and 105 μm meshes. The filtered acinar cells were added dropwise to 20 mL HBSS + 30% FBS, centrifuged at 1000 rpm for 2 min (4 °C), re-suspended in 10 mL of Waymouth complete media (1% FBS, 0.1 mg/mL trypsin inhibitor, 1 μg/mL dexamethasone), and infected with adenovirus harboring GFP or GFP;cre (Vector Biolabs, Malvern, PA) for 3–5 h at 37 °C. For 3D culture, cell culture plates were coated with collagen I diluted in Waymouth media without supplements, and freshly isolated pan-creatic acinar cells were added on top, and overlayed with Waymouth complete media. On day 5, cells were harvested and analyzed as described in the figure legend.

### DAB Immunohistochemistry and trichrome staining of tissues

Samples were deparaffinized for 1 h at 60 °C, de-waxed in xylene (five times for 4 min), and gradually re-hydrated with ethanol (100%, 95%, 75%, twice with each concentration for 3 min). The rehydrated tissue samples were rinsed in water and subjected to antigen retrieval in citrate buffer (pH 6.0). Samples were treated with 3% hydrogen per-oxide for 5 min to reduce endogenous peroxidase activity, washed with PBS containing 0.5% Tween 20, and blocked with protein block serum-free solution (DAKO) for 5 min at room temperature. Primary antibodies were diluted (Supplementary Table 1) in antibody diluent

background reducing solution (DAKO) and visualized using the EnVision Plus Anti-Rabbit Labeled Polymer Kit (DAKO). H&E staining was performed as previously described[40,45]. Images were captured using the Aperio AT2 scanner (Leica Biosystems, Buffalo Grove, IL) and ImageScope software. For trichrome staining, tissue slides were stained using the Masson trichrome stain kit (Sigma-Aldrich).

## In situ hybridization (ISH) methods

*For ISH with CVB3 probe1*: Paraffin-embedded sections were deparaffinized and digested with 8 mg/mL pepsin in 0.2 N HCl (Dako, Carpinteria, CA) for 10 min. The slides were then washed in water for 1 min, 100% ethanol for 1 min, and air-dried. The in situ hybridization was performed using a 3′-biotin labeled probe (2 pmol) specific for the CVB3 capsid protein (5′-TCCAGGGTATACACAGCACGCAACTTGAT TGTAGCCCCAC-3′; 100% sequence homology to multiple CVB3 isolates/variants including Woodruff, Macocy, Nancy, MKP, RD, LRY007, NSW-V13B-2008 and many others), diluted in in situ hybridization buffer (Enzo Life Sciences, Farmingdale, NY). After a brief 5 min denaturation step at 85 °C, the samples were incubated for 16 h at 37 °C. Following hybridization, the slides were washed with 0.2× SCC buffer (30 mM NaCl, 3 mM sodium citrate, pH 7) for 10 min, briefly dipped in water, and then incubated for 2 h at room temperature with anti-streptavidin-HP (1:50, R&D Systems, Minneapolis, MN) diluted in PBS. After 3 washes with PBS, mRNA was detected using 3,3′-Diaminobenzidine (DAB; from Dako), followed by hematoxylin staining. Images of the slides were captured using the Aperio AT2 scanner (Leica Biosystems).

In-situ hybridization with CVB3 probe 2 was done using RNA-scope® Assay 2.5 HD Reagent Kit-Brown (Advanced Cell Diagnostics [ACD], Hayward, CA) as per the manufacturer's protocol, with some minor adjustments. In summary, FFPE 5 μm sections were deparaffinized and dried in a desiccator at room temperature (RT) until the next day. Using the kit reagents, the slides were incubated with hydrogen peroxide for 10 min, boiled in target retrieval solution for 8 min (modification for the pancreas), and treated with protease plus for 15 min at 40 °C. RNA probe V-CVB3 (ACD #409291) was added for 2 h (40 °C), followed by six amplification steps (Amp 5 incubation time extended to 1 h), with 2 min washes in between the steps. The slides were then incubated with DAB for signal detection, counterstained with hematoxylin, dehydrated in ethanol and xylene, and coverslipped. The slides were scanned using ScanScope XT scanner and analyzed using ImageScope software (Aperio, Vista, CA).

## Senescence assay

Cryosections of fresh frozen mouse pancreas (OCT embedded) were stained for β-galactosidase activity using the Senescence β-Gal Staining kit (#9860) from Cell Signaling according to the manufacturer's protocol, with some modifications. Briefly, 10 μm cryosections were fixed for 15 min at room temperature using 1× Fixative Solution (kit), rinsed with 1x PBS two times and incubated with 1× β-Gal Staining Solution (kit) for 48 h at 37 °C in a sealed box in an incubator (no $CO_2$). Slides were then rinsed with 1× PBS three times and counterstained with Nuclear Fast Red for 5 min.

## Proximity ligation assay (PLA)

PLAs were performed using the Duolink In Situ Detection Reagents Red from Sigma-Aldrich, according to their protocol. The dilutions for the antibodies used are shown in Supplementary Table 1.

## RNA-seq

RNA from microdissected tissue was isolated using RNeasy Micro Kit (Qiagen, Hilden, Germany), followed by a quality analysis using Agilent 2100 Bioanalyzer (Agilent, Santa Clara, CA). The samples were then sequenced using Illumina TruSeq Stranded Total RNA (Degraded) Library Prep at the Mayo Clinic Gene Expression Core.

## Quantitative PCR and RT-PCR analyses

For quantitative PCR, cellular RNA was isolated using Exiqon miRCURY RNA isolation kit (Woburn, MA) according to the manufacturer's instructions. Equal amounts of total RNA were converted to cDNA using the High Capacity cDNA RT Kit (Applied Biosystems, Bedford, MA). Quantitative PCR was performed using the QuantStudio 7 Flex Real-Time PCR System (Applied Biosystems), using the TaqMan Fast Mix 2x (Applied Biosystems) and below primer sets. The thermocycler program was as follows: 95 °C for 20 s, 40 cycles of 95 °C for 1 s, and 60 °C for 20 s. Probe/primer sets were purchased from Applied Biosystems (Mm99999915_g1 for GAPDH, Mm00438355_m1 for *Cxadr*/CAR). Amplification data were collected by a Prism 7900 sequence detector and analyzed with Sequence Detection System software (Applied Biosystems). Data were normalized to mouse GAPDH, and mRNA abundance was calculated using the $\Delta\Delta C_T$ method.

## ELISA

Immulon 2HB flat bottom microtiter plates (Thermo Scientific) were coated with 0.5 μg of VP1-1/CVB3 peptide[47] dissolved in a 50 mM carbonate/bicarbonate buffer at pH 9.6. Mouse IgG standards (Sigma Aldrich) were serially diluted in the same buffer starting from 500 ng/mL, and plates were incubated overnight at 4 °C. Plates were blocked with 200 μL of blocking buffer (1% BSA in PBS) for 2 h at room temperature and washed with wash buffer (0.05% Tween-20 in PBS). Serum was collected the same day, diluted 1:50 in blocking buffer, added to the plates and incubated overnight. The next day, plates were washed and incubated with HRP-conjugated secondary antibody (Millipore) for 1 h at room temperature and washed again. Plates were developed by adding TMB substrate (Thermo Scientific) for 1 min and stopping the reaction with 1 N HCl. Signal was detected by reading OD at 450 nm wavelength using a BioTek Synergy HT Plate Reader.

## Statistics and reproducibility

Statistical analyses were completed in GraphPad Prism (GraphPad Inc., La Jolla, CA) and tests used are noted in the figure legends. $p < 0.05$ was considered statistically significant. If any data was excluded, it was done so using the ROUT method for outlier detection in GraphPad Prism with $Q = 1\%$, and this is indicated in the figure legends. All experiments (cell biological and biochemical) shown have been performed as biological replicates 3 times. For quantification analyses in all animal experiments, pancreatic samples from $n = 3$ up to $n = 6$ mice (stated in the figure legends) have been used. Mice were randomly assigned to treatment groups with approximately equal numbers of males and females in each treatment. No statistical method was used to predetermine the sample size. Investigators were not blinded to allocation during experiments and outcome assessment. IHC data was quantified by manual counting of positive cells or by using the Aperio Positive Pixel Count Algorithm. Error bars represent ±standard deviation (SD).

## Reporting summary

Further information on research design is available in the Nature Portfolio Reporting Summary linked to this article.

# Data availability

The RNAseq data generated in this study have been deposited in the Gene Expression Omnibus (GEO) database under accession code GSE280352. Source data are provided with this paper.

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

## Acknowledgements

This work was supported by the NIH grants R01CA200572 (P.S.), R01CA229560 (P.S.), R01DK139604 (P.S.), R50CA211462 (P.M.G.), R01HL111938 (D.F.), and P50CA127297 (M.A.H.). Additional support to P.S. was from a grant from the Chartrand Foundation, the Champions For Hope (Funk-Zitiello Foundation), and funding from the Mayo Clinic Center for Individualized Medicine (CIM) Clinomics Translational Program. This manuscript is dedicated to Judi Zitiello who lost her 11-year battle with pancreatic cancer in 2024.

## Author contributions

Conceived and designed the experiments: L.I.B., X.L., V.P., and P.S. Performed the experiments: L.I.B., X.L., V.P., H.D., A.K.F., B.H.E., and D.F. Analyzed the data: L.I.B., X.L., H.D., A.K.F., V.P., E.A.T., L.Z., and P.S. Provided tools or tissue samples: M.A.H., P.M.G., M.K., and D.C. Wrote the paper: P.S.

## Competing interests

The authors declare no competing interests.
