## [Transparent Peer Review file · Nature Communications]

Coxsackievirus and adenovirus receptor expression facilitates enteroviral infections to drive the development of pancreatic cancer

Corresponding Author: Dr Peter Storz

Version 0:

Reviewer comments:

Reviewer #2

(Remarks to the Author)

The authors have thoroughly and meaningfully responded to my comments from the last submission, as well as those of referee #1.

Reviewer #4

(Remarks to the Author)

Bastea et al.

Nat Commun

Manuscript #NCOMMS-24-64522-T

The authors have made a good-faith effort to address the concerns of the first-round reviewers. I have just two simple requests, one related to reporting and one related to a missing control for the new experimental results added to the revision:

1) The reporting summary indicated the mice of both sexes were randomly assigned to the treatment or vehicle groups, because there is no sex dependence of pancreas cancer. This is true, but BL/6 animals respond differently to CVB3 infection by sex (PMID 38750778). If the randomization was relatively balanced, then there should be no issues with the fundamental conclusions, but I request that the individual replicates have the markers changed to indicate the sex of each replicate (e.g., Fig. 3G,H).

2) The new results with CxadrF/F animals are very encouraging, but nowhere could I find evidence that CVB3 infection was prevented in p48cre ;LSL-KrasG12D;CAR-/- mice. This could be done by IHC or CISH as in Fig. 2 or by qPCR as in Extended data Fig. 3B.

Response NCOMMS-24-64522-T

We would like to thank both reviewers for reevaluating our revised manuscript. In the current manuscript (revision 2) we have addressed the two points raised by reviewer 4, and have included two additional Supplemental Data figures (S4 and S8B), but also modified figures 4B, 4C and 6B to indicate sex of the mice used. Below is a point-to-point response.

Response to Reviewer #1:

We would like to thank this reviewer for re-evaluating our revision and for her/his comment that ***“The authors have thoroughly and meaningfully responded to my comments from the last submission, as well as those of referee #1”***.

Response to Reviewer #4:

We would like to thank this reviewer for pointing out that ***“The authors have made a good-faith effort to address the concerns of the first-round reviewers”***. This reviewer had two requests that are addressed in the response below.

1) ***“The reporting summary indicated the mice of both sexes were randomly assigned to the treatment or vehicle groups, because there is no sex dependence of pancreas cancer. This is true, but BL/6 animals respond differently to CVB3 infection by sex (PMID 38750778). If the randomization was relatively balanced, then there should be no issues with the fundamental conclusions, but I request that the individual replicates have the markers changed to indicate the sex of each replicate (e.g., Fig. 3G,H).”*** – This is an excellent point. In our revised manuscript, we have indicated the sex of animals in each group for Figure 3D (included as Supplemental Data S4) and in Figures 4B, 4C and 6B.

New Supplemental Data S4:

New Figures 4B and 4C:

New Figure 6B:

In conclusion, we do not see obvious sex differences such as that male mice would be more susceptible (this becomes most evident in the survival curve shown in Figure 6B). According to our co-author Dr. Fairweather, whose laboratory has a long-term focus on sex differences of CVB3 infection-induced myocarditis, a relatively large number of mice (similar as shown PMID 38750778 and for other myocarditis studies) would be needed to determine a potential increased susceptibility of male or female mice for pancreatic cancer formation in KC mice. This important aspect needs to be addressed in future studies.

We have included the above cited reference into our manuscript and briefly discuss potential sex differences due to CVB3 infections susceptibility on pages 11/12. It reads: “Of note, within

these experimental groups female or male animals were randomly assigned (indicated by symbols). While in some CVB3-induced myocarditis studies male mice may have a slightly-increased susceptibility to infection and severity of the effects ⁴¹, we did not observe any obvious effects on lesion progression (Supplemental Data S4, Fig. 4B) and survival (Fig. 6B) due to sex of the animal. However, to determine if CVB3 in KC mice could have effects based on sex, animal numbers would need to be significantly increased in future studies.”

2) **“The new results with *CxadrF/F* animals are very encouraging, but nowhere could I find evidence that CVB3 infection was prevented in *p48^{cre};LSL-Kras^{G12D};CAR^{-/-}* mice. This could be done by IHC or CISH as in Fig. 2 or by qPCR as in Extended data Fig. 3B.”**

– This is an excellent point, and we have included additional supplemental data (**Supplemental Data S8B**), in which we have used IHC to confirm that CVB3 cannot be detected in KC;*CAR^{-/-}* mice at the endpoint of this experiment.

New Supplemental Data S8B: